METHODS AND RESOURCES

# Exploring bacterial diversity via a curated and searchable snapshot of archived DNA sequences

**Grace A. Blackwell**[1,2], **Martin Hunt**[1,3], **Kerri M. Malone**[1], **Leandro Lima**[1], **Gal Horesh**[2¤], **Blaise T. F. Alako**[1], **Nicholas R. Thomson**[2,4‡], **Zamin Iqbal**[1‡*]

1 EMBL-EBI, Wellcome Genome Campus, Hinxton, United Kingdom, 2 Wellcome Sanger Institute, Wellcome Genome Campus, Hinxton, United Kingdom, 3 Nuffield Department of Medicine, University of Oxford, Oxford, United Kingdom, 4 London School of Hygiene & Tropical Medicine, London, United Kingdom

¤ Current address: Chesterford Research Park, Cambridge, United Kingdom
‡ These authors are joint senior authors on this work.
* zi@ebi.ac.uk (ZI); nrt@sanger.ac.uk (NRT)

**Data Availability Statement:** The 661,405 assemblies as well as the COBS, minHash and pp_sketch indices are available from: ftp.ebi. ac.uk/pub/databases/ENA2018-bacteria-661k. In

## Abstract

The open sharing of genomic data provides an incredibly rich resource for the study of bacterial evolution and function and even anthropogenic activities such as the widespread use of antimicrobials. However, these data consist of genomes assembled with different tools and levels of quality checking, and of large volumes of completely unprocessed raw sequence data. In both cases, considerable computational effort is required before biological questions can be addressed. Here, we assembled and characterised 661,405 bacterial genomes retrieved from the European Nucleotide Archive (ENA) in November of 2018 using a uniform standardised approach. Of these, 311,006 did not previously have an assembly. We produced a searchable COmpact Bit-sliced Signature (COBS) index, facilitating the easy interrogation of the entire dataset for a specific sequence (e.g., gene, mutation, or plasmid). Additional MinHash and pp-sketch indices support genome-wide comparisons and estimations of genomic distance. Combined, this resource will allow data to be easily subset and searched, phylogenetic relationships between genomes to be quickly elucidated, and hypotheses rapidly generated and tested. We believe that this combination of uniform processing and variety of search/filter functionalities will make this a resource of very wide utility. In terms of diversity within the data, a breakdown of the 639,981 high-quality genomes emphasised the uneven species composition of the ENA/public databases, with just 20 of the total 2,336 species making up 90% of the genomes. The overrepresented species tend to be acute/common human pathogens, aligning with research priorities at different levels from individual interests to funding bodies and national and global public health agencies.

## Introduction

The widespread availability of high-throughput sequencing has resulted in a huge wealth of bacterial genomic data collected from countries all over the world that are shared openly

addition, the 661,405 assemblies have been deposited as third-party assemblies to the European Nucleotide Archive (ENA) under Umbrella project PRJEB46036 (individual project accessions for the major species can be found in S1 Table). The pipeline used for download and assembly of reads from the ENA https://github.com/iqbal-lab-org/assemble-all-ena. Additional metadata, characterisation files and Rnotebooks detailing figure generation have been deposited in figshare (https://dx.doi.org/10.6084/m9.figshare.16437939): -Full metadata downloaded from the ENA for each assembly in json form (Json1_ENA_metadata) -Full QC and general characterisation including AMR gene and plasmid replicon detection, for each assembly in json form (Json2_QC_characterisation_amr_plasmid) -Kraken/Bracken output including the top 50 species for each assembly (File1_full_krakenbracken) -The taxid lineage of the major species determined using NCBITaxa (File2_taxid_lineage_661K) -Summarised metadata from the ENA for each assembly (File3_metadata_661K) -Summarised QC and general characterisation for each assembly (File4_QC_characterisation_661K) -Summarised AMR genes, MCR status, plus genes, plasmid replicons for each assembly (File5_AMR_plasmids_661K) -Presence/absence matrix of AMR genes in each assembly (File6_AMR_presenceabsence_661K) -Class of each AMR gene extracted by AMRFinder (File7_gene_class_AMRFinder) -Presence/absence matrix of plasmid replicons in each assembly (File8_plasmidreplicons_presenceabsence_661K) -All available metadata from the ENA for each sample in the 661K (File9_all_metadata_ena_661K.txt) -Bacterial assemblies in GenBank as of the 6th March 2021 (File10_Genbank_assemblies_06.03.21.txt) -Bacterial assemblies in PATRIC as of the 6th March 2021 (File10_PATRIC_assemblies_06.03.21.txt) -Code used to generate figures in the QC and filtering section (Rnotebook1_QC_filtering_section) -Code used to generate figures in the Species breakdown section (Rnotebook2_species_breakdown_section) -Code used to generate figures in the AMR section (Rnotebook3_AMR_section_figures) -Code used to generate the figures comparing the composition of the 661K, GenBank assemblies and PATRIC database (Rnotebook4_661K_vs_genbank_patric).

**Funding:** This work was supported by Wellcome Trust (206194). M.H. was funded by a Wellcome Trust/Newton Fund-MRC Collaborative Award (200205) and an award from the Bill & Melinda

through the public archives, representing a unique and essential resource. Studying the extreme diversity of bacterial species is of broad interest to communities working in basic science, agriculture, and medicine. Beyond their primary function of genomic data storage, sequence repositories show trends in funding, biases in the collection strategies of bacteria, and even reveal the drive and focus of individuals pursuing particular lines of research. Sequence read data are held by members of the International Nucleotide Sequence Database Collaboration (INSDC) [1], who include the DNA Data Bank of Japan (DDBJ), the European Bioinformatics Institute (EMBL-EBI), and the National Centre for Biotechnology Information (NCBI). Submission of genomic data to the European Nucleotide Archive (ENA) (EMBL-EBI) or its INSDC partners (DDBJ Sequence Read Archive, DRA, for DDBJ and Sequence Read Archive, SRA, for NCBI) has become a central and mandatory step in the dissemination of research to the scientific community and a way to ensure open and free access to data [1]. Each of these repositories host the raw read data as well as genome assemblies, at different levels of completeness, which have been submitted by a user. These archives are continuing to grow at a remarkable rate, with current estimation of doubling time of datasets in the ENA to be just over 2 years (https://www.ebi.ac.uk/ena/browser/about/statistics). The ever increasing data size presents difficulties for storage capacity. A general user's ability to access and effectively use the data is restricted, whether due to their computational skills, the biological question, the volume of data, the IT infrastructure, or other resources required. Furthermore, once a user has their genomes of interest, significant processing for quality control, including remapping of the underlying raw data, is ideally required prior to applying specific analyses. Even after that is done, any potential discovery needs to be carefully reviewed in the light of the fact that different genome assemblers were applied to each genome [2], and, realistically, it is very hard to rule this out with confidence. Careful users therefore might assemble the raw data from scratch, and, over time, many of these processing steps will be performed repeatedly by different researchers worldwide.

Other databases exist that provide a high level of curation, including NCBI's RefSeq [3]. RefSeq (195,316 assemblies in September 2020) is composed of a selection of assemblies that have been submitted to INSDC databases that meet their quality control requirements, and most have been reannotated using NCBI's prokaryotic genome annotation pipeline [4] to provide consistency across the data. The assemblies are widely used for taxonomic identification [5,6], but are also commonly used to examine the distribution of genes or elements of interest or as test sets for new algorithms or programmes [7,8]. However, the RefSeq assemblies have been collated progressively over time using a range of assembly algorithms, making the assemblies less consistent and so potentially more problematic for drawing wide-ranging conclusions [9,10].

Attempts to standardise the assembled dataset tend to have a community focus such as Enterobase, which holds sequencing data from the Enterobacteriaceae, and includes curated genome data for 466,670 *Salmonella*, *Escherichia/Shigella*, *Clostridioides*, *Vibrio*, *Helicobacter*, *Yersinia*, and *Moraxella* genomes [11]. Enterobase gathers sequence data with associated metadata by actively searching for new sequence submissions for supported genera or through direct submissions. The raw data are then processed in a uniform way (assembly and annotation), and basic organism-specific typing is performed [11]. However, while standardised, the scope of this type of database is by definition limited. Depending on an individual's focus, this can act to further fragment genome data and lead to even more incompatibility issues if the complete genome dataset, agnostic of organism, is to be analysed.

The above databases lack comprehensive facilities for obtaining subsets of interest matching different criteria, and the support of a few key use cases would significantly transform their utility. First, it should be possible to filter for genomes containing a given DNA sequence (e.g.,

Gates Foundation Trust (OPP1133541). The funders had no role in study design, data collection and analysis, decision to publish, or preparation of the manuscript.

**Competing interests:** The authors have declared that no competing interests exist.

**Abbreviations:** AMR, antimicrobial resistance; COBS, COmpact Bit-sliced Signature; DDBJ, DNA Data Bank of Japan; DRA, DDBJ, Sequence Read Archive; EMBL-EBI, European Bioinformatics Institute; ENA, European Nucleotide Archive; GTB, Genome Taxonomy Database; INSDC, International Nucleotide Sequence Database Collaboration; MCR, multi-class resistant; NCBI, National Center for Biotechnology Information; SNP, single nucleotide polymorphism; SRA, Sequence Read Archive; WGS, whole genome sequencing.

a specific single nucleotide polymorphism (SNP), gene, or plasmid). We know that there is currently no known algorithm that allows a full BLAST alignment on this scale, but it has been shown that *k*-mer indexes allow presence/absence search for SNPs, genes, and plasmids [12]. Secondly, it would be valuable to be able to select a subset that is representative of diversity of a species, genus, or other taxon—this would be enabled by precalculating genetic distances or a phylogenetic tree. Finally, availability of lightweight methods for whole genome comparison [7,13] would enable users to easily query the database for any genomes similar to their own. With these, a host of possibilities can be realised including the ability to easily perform generic queries of a user such as "Has this plasmid isolated from *Salmonella enterica* ever been seen in other species?," "Does this SNP occur in many *Escherichia coli* genomes?," and "How do I get a set of *Listeria* genomes representative of the genomic diversity?."

Here, we present a uniformly processed archive of 661K bacterial genomes that were available in the ENA at the end of November in 2018. These include 311,006 isolates that did not previously have publicly available assemblies. Through the quality control steps, characterisation of the assemblies, and the provision of multiple search tools, we remove technical barriers for the interrogation of the public sequences. We use these data to examine the composition of the sequencing archives and in doing so highlight the influence of sampling and sequencing trends.

## Results

### Construction of a unified resource

On the November 26, 2018, there were 880,947 bacterial read sets available in the ENA. Those that were single ended or were sequenced on the PacBio or Nanopore platform were removed, and the 710,696 unique sample accessions were submitted to an assembly pipeline (see Methods), yielding 664,877 assemblies. A subset of these (3,472 assemblies) had a genome length significantly outside that expected of a bacterial organism (smaller than 100 kb or larger than 15 Mb), leaving 661,405 standardised assemblies. Quality control and general characterisation were performed on these 661K assemblies (see Methods). Standard quality control cutoffs, many of which are consistent with the threshold for inclusion for RefSeq, were applied to identify genomes that were of high assembly quality (S1 Fig). These assemblies represent complete or almost complete genomes that were not overly fragmented and had a genome length within an acceptable deviation of that expected of its species (see Methods). A total of 639,981 assemblies reached or exceeded these thresholds (S2A Fig, filter status 4). The metadata available for each genome from the ENA as well as quality control and characterisation results have been collated and are available on Figshare.

Analysis of the read sets contributing to each assembly using Kraken 2 [6] and then refining the output using Bracken [14] revealed that for 94.1% (602,406/639,981) of the assemblies, the major taxonomic species accounted for 90% or greater of the total reads in that set (S2C Fig). Hence, there was little evidence of mixed samples or significant contamination. Importantly, lowest common ancestor approaches for species assignment, such as Kraken 2, are not ideal if the major taxon is a member of a species complex. Therefore, we calculated an adjusted abundance (see Methods) for members of the *Mycobacterium tuberculosis* complex, *Bacillus cereus* sensu lato group, or where genera or species represent taxonomic anomalies such as the division of *Shigella* sp. and *E. coli*, which is based on clinical imperative rather than a true taxonomic distinction [15,16]. For some species, including *Burkholderia pseudomallei*, *Bordetella pertussis*, *Mycobacterium ulcerans*, and *Campylobacter helveticus*, the major species abundance in more than 97.6% of their assemblies were less than 90% using these approaches (S3 Fig), despite passing earlier quality control thresholds for

contamination (S2D Fig). This indicates that there are likely limitations with the methods for species identification used here. Of note, 91.7% (606,508) of the assemblies in the 661K had been submitted with species metadata that was consistent with the major species we identified *in silico* from sequence.

To facilitate access and usage, we have added 3 indices that can be downloaded along with the 661,405 assemblies. The COmpact Bit-sliced Signature (COBS) index [17] allows the user to search for sequence by breaking it into constituent *k*-mers. This methodology has previously been shown to work for SNPs, whole genes, or even extrachromosomal elements such as plasmids [12,17]. Secondly, the MinHash index [13], containing signatures of the assemblies, can be used to search for matches to any query genomes (i.e., to find similar genomes). A third index, constructed using the library sketching function of PopPUNK [18], includes the calculated core and accessory distances between the 661K assemblies. Genetic distance estimates for any subset of assemblies can be extracted quickly and easily from this index.

The sample accessions of the 661,405 assemblies were compared to those in NCBI Bacterial Assemblies (867,940 accessions, March 6, 2021) [19] and the PATRIC database (422,590 accessions, March 6, 2021) [20]. The corresponding upset plot (S4A Fig) revealed that 311,006 sample accessions were unique to the 661,405 assemblies. The top 50 species represented in these previously unavailable genomes are shown in S4B Fig. In addition to producing assemblies that are uniform and searchable via various means, this resource has substantially increased the availability of sequences that were previously only in the read archives.

## Diversity and sequencing trends

The 639,981 high-quality assembled genomes comprised 2,336 species (S2B Fig), and the breakdown of the genomes based on the year that they were made public in the ENA is shown in S5A Fig. Despite the considerable number of species in this dataset, sampling was extremely unevenly distributed, with just 20 species accounting for 90.6% of the assembled dataset (Fig 1A). Within this, *S. enterica* accounted for almost a third of the data (28.0%), while *E. coli* (13.4%), *Streptococcus pneumoniae* (7.9%), *Staphylococcus aureus* (7.4%), and *M. tuberculosis* (7.3%) combined constituted over 35% (Fig 1A). The final 9.4% of the assemblies comprised 2,315 species, i.e., 99.1% of the species diversity (Fig 1A and 1B). A similar trend is revealed when the contributing sequencing projects are examined, with 50% of the data originating from 50 sequencing projects (S5B Fig), a small fraction of the total 23,316 projects. The majority of the sequencing projects (20,002) only yielded a single assembly. Unsurprisingly, 3 of the 5 largest projects focus on *S. enterica*. These include the PulseNet *S. enterica* genome sequencing project (PRJNA230403, 59,011 assemblies, 2014 onwards) run by the Centre for Disease Control [21], the Salmonella Reference Service (Gastrointestinal Bacteria Reference Unit) from Public Health England (PRJNA248792, 35,942 assemblies, 2014 onwards) [22], and the GenomeTrakr project (PRJNA186035, 19,418 assemblies, 2012 onwards) run by the United States Food and Drug Administration Center for Food Safety and Applied Nutrition [23]. The ramping up of these large public genomic surveillance projects in 2014 contributed to *S. enterica* dominating as the major bacterium sequenced from 2015 (Fig 1C, S5C Fig). The Global Pneumococcal Sequencing GPS study I (PRJEB3084, 20,667 assemblies), which focuses on *S. pneumoniae* [24,25], and a US public health project focusing on *E. coli* and *Shigella* (PRJNA218110, 20,508 assemblies, 2014 onwards) [21] are the third and fourth largest projects in the archive. Specific interests of individuals or groups have also contributed to these sequencing trends, although the impact is more obvious in the earlier years, where organisms such as *B. pertussis* (PRJEB2274) and *Salmonella bongori* (PRJEB2272) were prominent but were overshadowed in later years (Fig 1C).

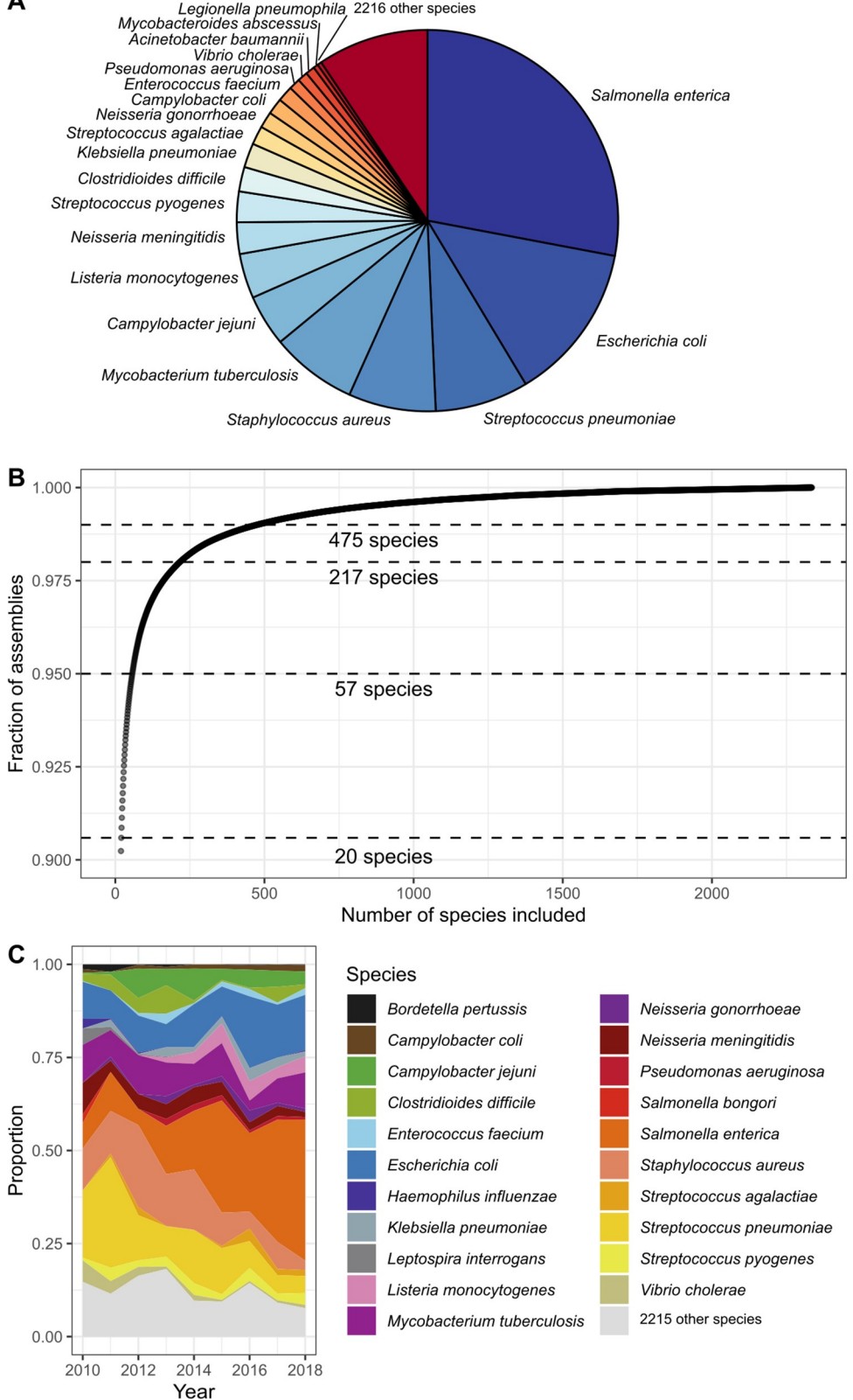

**Fig 1. Species composition of the 639,981 high-quality assemblies. (A)** Relative proportions of species to the data as a pie chart. Note that 90% of the assemblies are from 20 bacterial species. **(B)** Fraction of assemblies covered by

accumulating bacterial species. **(C)** Tracking proportions of the top 10 bacterial species for each year. The data underlying this figure may be found in https://doi.org/10.6084/m9.figshare.16437939.

## Distribution of and accumulation of antimicrobial resistance genes

One of the major selective forces that has perturbed bacterial populations has been the development and widespread therapeutic use of antimicrobials since the 1940s [26–28]. Antimicrobial resistance (AMR) is highlighted as one of the greatest threats to human health [29]. It has been estimated that if no action is taken, 10 million people worldwide could die from drug-resistant infections each year by 2050 [30]. We have genotypically predicted the presence of AMR, virulence, and stress response genes for all assembled genomes (see Methods), but the results shown below are for the 602,407 high-quality genomes with a confident major species (>90% abundance major species), unless specified otherwise. Our approach detects both genes that are core to a species, usually located on the chromosome(s), as well as those that have been horizontally acquired and are chromosomally located or otherwise located in extrachromosomal elements, such as plasmids. However, specific point mutations/deletions are not considered in this analysis.

In total, 1,655 known AMR gene variants were identified. Gene variants showed different distribution ranges across the assembled taxa, with 135 gene variants detected in 2 or more phyla. This reduced to just 73 when a stricter 98% threshold for abundance of the major species was set to limit the effects of low-level contamination commonly seen in submitted data (S6 Fig). Gene variants with more restricted distribution patterns, such as those found only within a particular genus or species, could represent variants that have recently arisen within that population, or were restricted directly, through, for example, gene expression, or indirectly based on the host range of the plasmid or vector that carries them. For example, the distribution patterns of the colistin resistance genes, first identified in 2016 [31], are at most detected within a bacterial order (*mcr-9*) or more commonly within a class (eg. *mcr-1*, *mcr-3*, and *mcr-5*), while some are only present in a single species (*mcr-1.7* and *mcr-4.1*).

An important trend seen in our data is the relative number of genomes carrying multiple AMR genes. The count of AMR genes in each genome for 2 of the most represented orders—Bacilli and Gammaproteobacteria—are shown in Fig 2. Most genera within the Bacilli contain genomes with fewer than 10 AMR genes. Some genomes belonging to the genera *Bacillus* and *Streptococcus* possess up to 10 or 11 resistance genes, while those from *Enterococcus* and *Staphylococcus* can carry up to 23 and 25 resistance genes in a single genome, respectively (Fig 2A). It is important to note that some of these resistance genes are core to a species (genes found in >95% of the genomes belonging to that species). For example, 3 of the genes counted in *Enterococcus* (*aac(6')-Ii*, *msr*C, and *eatA*) were core, consistent with previous analysis [32,33]. Similarly in *S. aureus*, the *tet38* efflux pump [34] is a core gene.

Gammaproteobacteria represent a large proportion of the gram-negative pathogens with many of the genera in this class possessing high AMR gene counts (Fig 2B)—most notably, *Acinetobacter*, *Escherichia*, *Klebsiella*, *Pseudomonas*, and *Salmonella*. Indeed, a small number of *E. coli* and *Klebsiella pneumoniae* genomes contained over 30 different AMR genes concurrently (of which only 1 and 4 genes were species core genes, respectively).

The above genera with high AMR gene carriage (Fig 2) harbour species identified by WHO as priority pathogens for research and development into new antibiotics [29]. The different categories described by WHO (critical, high, and medium) are displayed in Fig 2, using the red, orange, and yellow triangles. Other genera, not on WHO priority list, show a high abundance of AMR genes, including *Vibrio*, *Citrobacter*, *Aeromonas*, and *Kluyvera*. Apart from

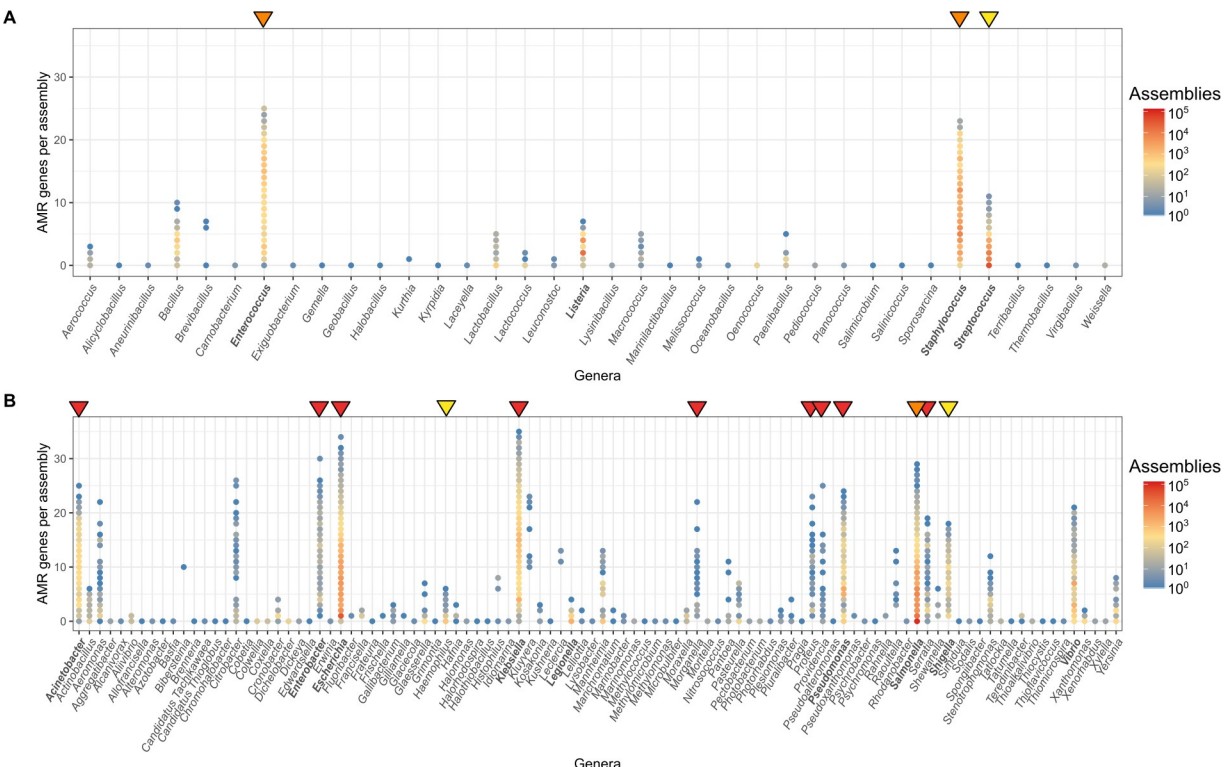

**Fig 2.** Number of AMR genes in individual genomes of the orders **(A)** Bacilli and **(B)** Gammaproteobacteria. Genera in bold contain species that are in the top 20 represented species in the 661K snapshot. Arrows above indicate genera that contain species that have been determined by WHO to be of critical (red), high (orange), and medium (yellow) priority pathogens for research and development into new antibiotics [29]. The Actinobacteria order is not shown as it does not contain a member of WHO priority pathogen list. The data underlying this figure may be found in https://doi.org/10.6084/m9.figshare.16437939. AMR, antimicrobial resistance.

*Vibrio*, these genera are not well represented in the collection. Greater surveillance of these organisms could, as it has done for the other priority organisms, reveal an increasingly resistant trend and stimulate research, essential for the design of rational AMR control strategies.

Further to examining the count of resistance genes in discrete genomes, we have predicted how many classes of antimicrobials the genes within a genome confers resistance to. We find that 35% of genomes (211,101/602,406) contain resistance to at least 3 classes of antimicrobials and have been defined here to be multi-class resistant (MCR). For a species to be described as MCR (red in Fig 3), at least half of the genomes from this species must be MCR (note this was only calculated for those species with at least 10 representatives). A total of 37 species were classed as MCR. WHO priority pathogens are well represented, although for *S. enterica* and *E. coli*, despite having some genomes conferring resistance to up to 12 and 14 different classes of antimicrobials respectively, the majority of samples are not MCR, although many may contain mutational resistance to antimicrobials such as fluoroquinolones. At the other end of the spectrum is *Enterobacter bugandensis*, where all 10 samples (from 3 different projects) contain genes conferring resistance to 8 classes of antimicrobials. *E. bugandensis* was only identified in 2016 and was associated with neonatal sepsis [35]. The species *Kluyvera intermedia* and *Vibrio cholerae*, in addition to possessing overall high numbers of AMR genes (Fig 3A), were also MCR. So too were the emerging opportunistic human pathogens *Raoultella planticola* [36] and *Corynebacterium striatum* [37] as well as the zoonotic pathogen *Histophilus somni* [38] and *M. tuberculosis*. However, the level of resistance in *M. tuberculosis* is likely to be

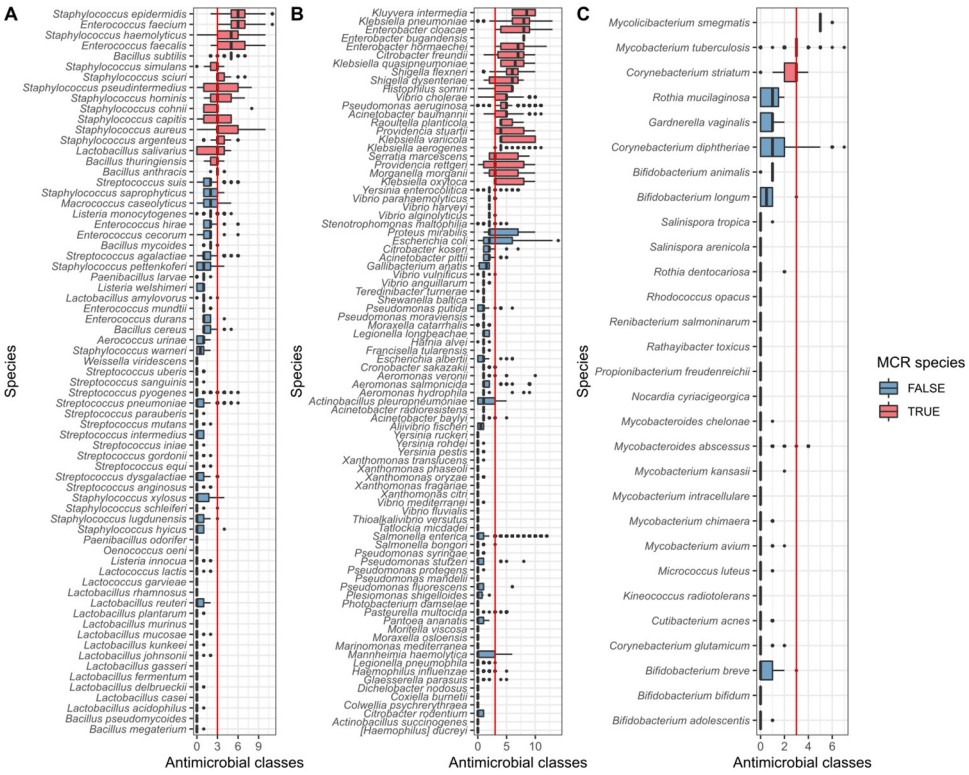

**Fig 3.** Predicted AMR profiles of species from **(A)** Bacilli, **(B)** Gammaproteobacteria, and **(C)** Actinobacteria, showing the number of predicted antimicrobial classes each isolate is resistant to, based on genetic profile. The red line indicates the threshold for MCR (predicted resistance to 3 classes of antimicrobials or more). Species are classed as MCR (red in figure) if at least 50% of the assemblies are MCR. Species included have at least 10 assemblies. The data underlying this figure may be found in https://doi.org/10.6084/m9.figshare.16437939. AMR, antimicrobial resistance; MCR, multi-class resistant.

underestimated here, as the main mechanism of resistance is through mutation [39], which is not considered.

## Discussion

Bacteria are a vast, diverse, and ancient family of single-celled organisms that dominate this planet. In our efforts to understand and categorise this most abundant life form, hundreds upon thousands of bacterial sequences are submitted yearly into sequence archives such as the ENA. In the last 2 decades and with the advent of cheap high-throughput short read sequencing, the trend has moved away from the submission of finished or draft genome assemblies to one where simply the raw reads are submitted to public archives. These data usually require substantial preprocessing before they are analysis ready. This takes significant time, expertise, and computational power to do. By uniformly processing the data present in the ENA in November of 2018, we have collated a set of 661,405 standardised assemblies, including 311,006 that were previously only available in the read archives.

The additional standard characterisation and quality control we have performed enables the data to be easily subsetted for the purposes of identifying all the assemblies of a particular species or sequence type or to those containing a specific AMR gene. Furthermore, this dataset can be interrogated for a specific gene or mutation using the COBS search index, for a specific genome by use of the provided MinHash index and glean estimations of genetic distances of

genomes of interest using the pp-sketch index. These facilities hint at the power of this unified resource, allowing phylogenetic relationships between genomes to be quickly elucidated and hypotheses rapidly tested. This resource will empower more scientists to harness the multitude of data in the ENA both for surveillance and public health projects, as well as to address questions of basic science.

The overrepresentation of species belonging to Proteobacteria, Actinobacteria, Firmicutes, or Bacteriodetes phyla is a trend consistent with reports of public database composition from 2002 [40] and 2015 [41]. Concerningly, there has been a reduction in the species diversity outside of these 4 phyla, decreasing from 29% in 2002, 8% in 2015, and 0.68% of all submissions analysed in the 661K snapshot. The count of 2,336 species in this snapshot is well below the number of bacterial species in the taxonomic databases such as NCBI taxonomy (>20,000 species, https://www.ncbi.nlm.nih.gov/Taxonomy/Browser/wwwtax.cgi?id=2) and Genome Taxonomy Database (GTB; >30,000 species, https://gtdb.ecogenomic.org/). Some of the sequence diversity within the snapshot may have been missed due to limitations of the Kraken 2 database used for taxonomic assignment and abundance estimation, a research project in its own right. For a small proportion of the assemblies (6.1%), a major species could not be assigned with high confidence, despite being shown by CheckM to contain little or no contamination, indicating that there was not a good match for it in the database (see Methods and S2D Fig). The inclusion of genomes originating from metagenomic sequences from different sources (e.g., gut, skin, soil, and ocean) would likely improve the overall species diversity, but the methods of assembly and analysis are very different to those used here.

Many of the sequenced genomes could be defined as MCR based on the carriage of AMR genes. While we observe many occurrences of AMR mechanisms in the 661K assemblies, both in the organisms that are already known to be problematic (species outlined on WHO priority pathogens list) and in newly emerged threats (such as *E. bugandensis*, *C. striatum*, and *R. planticola*), it is difficult to estimate how well these reflect the true prevalence of resistance in a given species. This is due to many biases, including, for example, how isolates were sampled, and some projects implementing preselection steps with only the antimicrobial resistant strains being then sequenced [42–44]. This intrinsically biases the archive, preventing prevalence estimations. It also limits the power to track the origins of accessory genes, and, consequently, the species interactions that can be inferred from this. Ideally, strategies to sequence a wider variety of species, including susceptible isolates, from diverse environments and global locations must be implemented before the dynamics of gene flow can be accurately studied.

The uniform resource of 661K bacterial assemblies that we present here removes several technical barriers to harnessing the wealth of public data stored in the ENA, enabling a broader community to access and leverage these data for their research. We envisage this to be a valuable resource that can provide the substrate for a wide range of future studies. Nevertheless, it is intrinsically limited through the nature of our scientific practice, by the diversity of sequences it holds. Rather, the current composition highlights the influences of the past quarter of century of funding and scientific focus. The enormous contribution of just a few projects shows that even the drive and focus of individual groups has influenced our view of recent bacterial diversity. Sampling and sequencing strategies must change if we want to reveal the bacterial "tree" of life.

## Methods

### Download of reads, assembly, and characterisation of genomes

The bacterial whole genome sequencing (WGS) datasets in the ENA as of November 26, 2018 were downloaded and assembled as a part of an assembly pipeline (https://github.com/iqbal-lab-org/assemble-all-ena) [45]. Only paired-end reads were included, and those where the

instrument platform was "PACBIO_SMRT" or "OXFORD_NANOPORE" were excluded. In addition, those with a library source of "METAGENOMIC" and "TRANSCRIPTOMIC" were also ignored. Available metadata and appropriate reads were downloaded, and if multiple read sets were available, they were appended together. Reads were assembled using Shovill v1.0.4 (Seeman, T., https://github.com/tseemann/shovill) with default options. Shovill uses SPAdes (v3.12.0) [46] for assembly and includes some additional pre- and post-processing steps that utilise Lighter [47], FLASH [48], Trimmomatic [49], SAMtools [50], BWA-MEM [50,51], seqtk (https://github.com/lh3/seqtk), Pilon [52], and samclip (Seeman, T., https://github.com/tseemann/samclip) to speed up the assembly and to correct minor assembly errors. A total of 664,877 assemblies were produced by this pipeline.

Separate from the assembly pipeline, Kraken 2 v2.0.8-beta [6] was run on the read FASTQ files using the Kraken 2 microbial database (2018, 30 GB), and the resulting taxonomy labels assigned by Kraken 2 were analysed by Bracken v2.5 [14] to estimate the species abundance within each set of reads. From the assemblies, contigs of less than 200 bp were removed using the script available at https://github.com/sanger-pathogens/Fastaq, and contigs of *k*-mer depth less than 10 were noted, but not removed. QUAST v5.0.2 [53] was used to summarise assembly statistics, and CheckM v1.1.2 [54] using the "--reduced_tree" flag was used for estimations of completeness and contamination of an assembly. Assemblies with a genome length of less than 100 kb or longer than 15 Mb were removed (3,472 assemblies), leaving 661,405 assemblies. A MinHash sketch of each assembly ("-n 5000") was produced using sourmash v3.5.0 [13]. A searchable *k*-mer database of the 661K assemblies was constructed by COBS (https://github.com/bingmann/cobs, checkout 7c030bb) using "compact-construct" with default options [17]. Core and accessory distances were calculated between the assemblies using poppunk_sketch v1.5.1 with default options except "--k-step 3" [18]. MLST was determined where possible using mlst v2.19.0 (Seeman, T., https://github.com/tseemann/mlst), *E. coli* phylotype determined using clermonTyping v1.4.1 [55], and *Salmonella* were serotyped using SeqSero2_package.py v1.1.1 [56]. Plasmid replicons were detected using Abricate v1.0.1 (Seeman, T., https://github.com/tseemann/abricate) with the plasmidfinder 2020-May-7 database [57], and AMR, heavy metal, and virulence genes were detected using AMRFinderPlus v3.6.15 [58], with standard thresholds of minimum identity (curated cutoff if it exists and 0.9 otherwise) and default coverage of 0.5. All figures were generated in R using ggplot2 v3.3.3 [59] and where required were edited manually using Inkscape 2 v0.92.

## Taxid lineage, species comparison, and adjustment species abundance

The taxid lineage of the major bracken species was acquired by NCBITaxa [60]. Where the major species from the Bracken analysis belonged to either the *M. tuberculosis* complex or *B. cereus* s.l. complex or was identified as a *Shigella* sp. or an *E. coli*, the remainder of the read assignments were examined to see if they belonged to other members of that complex. If they were members, their assigned percentage was added to that of the major species.

## High-quality assemblies

Filtering was applied using the reports generated by QUAST and CheckM analysis for each genome. The high-quality assemblies met the requirements of less than 2,000 contigs, a genome length that is within the acceptable range for that species, as calculated for species with 4 or more assemblies in GenBank (ftp://ftp.ncbi.nlm.nih.gov/genomes/ASSEMBLY_REPORTS/species_genome_size.txt.gz, August 27, 2020) or is unknown, a N50 of greater than 5,000, a completeness score of at least 90%, and a contamination score of less than or equal to 5%. In total, 639,981 assemblies met these requirements.

## Composition against other public databases

The list of genomes in NCBI assemblies was downloaded (https://ftp.ncbi.nlm.nih.gov/genomes/ASSEMBLY_REPORTS/assembly_summary_genbank.txt, March 6, 2021) [19], and the taxid lineage of each genome was acquired by NCBITaxa [60]. In total, 867,940 belonged to the bacterial superkingdom. The list of 422,590 genomes that were in PATRIC [20] (March 6, 2021) were also downloaded. The sample accessions in the 3 databases (661K snapshot, NCBI assemblies, and PATRIC) were compared using UpSetR v1.4.0.

## Multi-class resistance

MCR was defined as containing genes conferring resistance to at least 3 classes of antimicrobial (antimicrobial classes were extracted from the AMRFinderPlus output). Only species with at least 10 samples were included, and a species was classed as MCR if at least 50% of individual assemblies were MCR.

## Supporting information

**S1 Table. Projects under Umbrella project PRJEB46036.**
(DOCX)

**S1 Fig. Quality control measures used to filter genome assemblies. (A)** Distribution of number of contigs per assembly in the collection. A total of 1,766 assemblies had greater than 3,000 contigs. Red line: Assemblies with more than 2,000 contigs were filtered from the high-quality assemblies. **(B)** Distribution of the N50 of each of the assemblies in the collection and 26,142 had an N50 of greater than 500,000. Red line: assemblies with an N50 less then 5,000 were filtered from the high-quality assemblies. **(C)** Comparison of genome size of each assembly to that expected of its species. Where available, the genome size range accepted for each species was extracted from ftp://ftp.ncbi.nlm.nih.gov/genomes/ASSEMBLY_REPORTS/species_genome_size.txt.gz, downloaded August 27, 2020. Those genomes that are greater than or less than the expected length was filtered from the high-quality assemblies. **(D)** Correlation between the genome completeness and contamination percentages produced by CheckM for each assembly. A total of 1,785 assemblies had a contamination score greater then 100%. Red lines indicate cutoffs applied; bottom right corner are the high-quality genomes. The data underlying this figure may be found in https://doi.org/10.6084/m9.figshare.16437939.
(TIFF)

**S2 Fig. QC of the 661,405 assemblies.** Number of **(A)** assemblies and **(B)** species remaining following each stage of filtering. Status 1; removal of genomes with >2,000 contigs, status 2; removal of genomes with an N50 <5,000, status 3; removal of genomes with length outside the range expected for that species (note: if expected range is not known, the assemblies are kept), status 4; assemblies with a completeness score = >90% and with a contamination score = <5%. A total of 639,981 assemblies passed the 4 levels of filtering and are the high-quality genomes. **(C)** Distribution of high-quality assemblies (filtering status 4) with >50% abundance of major species. A total of 9,595 assemblies were below this threshold. **(D)** Within-sample abundance of major species vs completeness of the high-quality assemblies. For (C) and (D), the abundance of major species is the adjusted abundance values (see Methods). The data underlying this figure may be found in https://doi.org/10.6084/m9.figshare.16437939.
(TIFF)

**S3 Fig. High-quality assemblies that have a major species abundance of less than 90%.** The data underlying this figure may be found in https://doi.org/10.6084/m9.figshare.16437939. (TIFF)

**S4 Fig. Composition of the assemblies in the 661K snapshot compared to those in the GenBank bacterial and PATRIC databases. (A)** Upset plot shows the number of shared sample accessions (also called biosample) in the 661K snapshot, the GenBank bacterial, and PATRIC databases. Each column corresponds to an exclusive intersection that includes the elements denoted by the dark circles, but not of the others. **(B)** The top 50 species in the 311,600 sample accessions unique to the 661K snapshot. The data underlying this figure may be found in https://doi.org/10.6084/m9.figshare.16437939. (TIFF)

**S5 Fig. Composition of the 639,981 high-quality assemblies. (A)** Breakdown of assemblies by year first public in the ENA. **(B)** Fraction of assemblies covered by accumulating projects. **(C)** Tracking proportions of the top 10 bacterial species for a year. The data underlying this figure may be found in https://doi.org/10.6084/m9.figshare.16437939. ENA, European Nucleotide Archive. (TIFF)

**S6 Fig. Distribution of AMR gene alleles by antibiotic class.** Gene variants are coloured by their level of spread, from being detected in genomes from different phyla to only found in a single species. The top graph includes genomes with the major species being $> = 90\%$ abundance, and the lower graph is when this threshold was increased to $> = 98\%$ abundance. The data underlying this figure may be found in https://doi.org/10.6084/m9.figshare.16437939. AMR, antimicrobial resistance. (TIFF)

## Acknowledgments

We thank Alexandre Almeida, Kate Mellor, Alyce Taylor-Brown, and all other members of the Iqbal and Thomson research teams for their useful discussions and suggestions. We would also like to thank John Lees for his helpful guidance and support when creating the pp-sketch index of the 661K assemblies. We also thank Colman O'Cathail and Nadim Rahman for their help with accession of the assemblies in the ENA.

## Author Contributions

**Conceptualization:** Grace A. Blackwell, Nicholas R. Thomson, Zamin Iqbal.

**Formal analysis:** Grace A. Blackwell.

**Investigation:** Grace A. Blackwell, Martin Hunt, Kerri M. Malone, Blaise T. F. Alako.

**Methodology:** Grace A. Blackwell, Martin Hunt.

**Project administration:** Nicholas R. Thomson, Zamin Iqbal.

**Software:** Grace A. Blackwell, Martin Hunt, Leandro Lima.

**Supervision:** Nicholas R. Thomson, Zamin Iqbal.

**Visualization:** Grace A. Blackwell, Gal Horesh.

**Writing – original draft:** Grace A. Blackwell, Nicholas R. Thomson, Zamin Iqbal.

**Writing – review & editing:** Grace A. Blackwell, Martin Hunt, Kerri M. Malone, Gal Horesh, Nicholas R. Thomson, Zamin Iqbal.

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
