## [Editor Report · Decision Letter 0]

12 Mar 2021

Dear Dr. Blackwell, 

Thank you for submitting your manuscript entitled "Exploring bacterial diversity via a curated and searchable snapshot of archived DNA sequences" for consideration as a Methods and Resources by PLOS Biology.

Your manuscript has now been evaluated by the PLOS Biology editorial staff, as well as by an academic editor with relevant expertise, and I am writing to let you know that we would like to send your submission out for external peer review.

Please re-submit your manuscript within two working days, i.e. by Mar 14 2021 11:59PM.

Kind regards,

Paula

---

Associate Editor

PLOS Biology

---

## [Decision Letter · Decision Letter 1]

6 May 2021

Dear Dr. Blackwell,

Thank you very much for submitting your manuscript "Exploring bacterial diversity via a curated and searchable snapshot of archived DNA sequences" for consideration as a Methods and Resources at PLOS Biology. Your manuscript has been evaluated by the PLOS Biology editors, an Academic Editor with relevant expertise, and by several independent reviewers.

In light of the reviews (below), we will not be able to accept the current version of the manuscript, but we would welcome re-submission of a much-revised version that takes into account the reviewers' comments. We cannot make any decision about publication until we have seen the revised manuscript and your response to the reviewers' comments. Your revised manuscript is also likely to be sent for further evaluation by the reviewers.

In particular, reviewer #2 thinks that the article in its current form may leave a reader with an impression that this is yet another bacterial genome database as you have not sufficiently explained the pros and cons of using this resource as opposed to another database like NCBI, GTDB or PATRIC. This reviewer suggests to include an additional set of analyses using minHash distances at various thresholds to compare what proportion of the genomic space provided by ENA is not covered in other databases like GTDB and PATRIC, and vice versa. Please address also the rest of the reviewers' concerns.

We expect to receive your revised manuscript within 3 months. 

**IMPORTANT - SUBMITTING YOUR REVISION**

*Re-submission Checklist*

*Published Peer Review*

*PLOS Data Policy*

*Blot and Gel Data Policy*

Sincerely,

Paula

---

Paula Jauregui, PhD

Associate Editor

PLOS Biology

REVIEWS:

Reviewer #1: Computational epidemiology.

Reviewer #2: Evolution of human pathogens.

Reviewer #1: Blackwell et al. describe a curated resource of bacterial genomes that they have generated from ENA records. They process this data using a particular workflow with state-of-the art, appropriate methods and analyse it with respect to taxonomic composition and AMR gene occurrence. 

Overall, while the technical part is well done, it seems to me that more exciting things and a more-in depth analyses of individual questions might be learned from this data resource than what is presented now in the article. As their main conclusion, in the abstract the authors state: Whilst these archives are rich in data, considerable processing is required before biological questions can be addressed…. An analysis on this scale revealed the uneven species composition in the ENA/public databases, with just 20 of the total 2,336 species making up 90% of the genomes. The over-represented species tend to be acute/common human pathogens."

That most data in public genome databases comes from very few, cultivable bacteria, most of which are human pathogens is a well-known fact, see, for instance, https://pubmed.ncbi.nlm.nih.gov/11864374/ ; https://www.ncbi.nlm.nih.gov/pmc/articles/PMC4361730/;
https://pubmed.ncbi.nlm.nih.gov/18341921/

Furthermore, I do not think that it is as complicated or time-consuming to come to this conclusion as the authors think using other approaches. One could, for instance, go to the NCBI taxonomy site, select the filter „has genome sequences", display 5 levels and sort the results locally by number of genomes.

Reviewer #2: The article by Blackwell and colleagues describes a novel resource: a large database of >660k bacterial genomes created from short-read data downloaded from European Nucleotide Archive in 2018. The genomes underwent a consistent assembly pipeline with an extensive quality control and a number of extra post-processing steps, including kmer indexing and minhash sketching. In addition, the authors have provided various analyses aimed at describing the genetic diversity, sequencing trends and the distribution of antimicrobial resistance genetic markers.

I find the article well written, methodology sound and the analyses interesting. Most importantly, the study offers both an exceptional new methodology and a fantastic resource for the entire field of biology. In my opinion the article deserves to be published in PLoS Biology. 

My major and only reservation is that the article in its current form may leave a reader with an impression that this is yet another bacterial genome database as the authors have not sufficiently explained the pros and cons of using this resource as opposed to another database like NCBI, GTDB or PATRIC. In fact the authors have stated directly that both NCBI and GTDB contain an order of magnitude greater number of bacterial species than the ENA archive. However, it is unclear whether the large number of assemblies provided with this resource is due to oversampling of major epidemiological lineages or due to inclusion of multiple novel lineages of the species already present. The ENA archive could also contain some species which are not present in other databases. To clarify this, I would suggest to include an additional set of analyses using minHash distances at various thresholds to compare what proportion of the genomic space provided by ENA is not covered in other databases like GTDB and PATRIC, and vice versa. A large increase in the number of genomes of major sequence types (eg, ST131 in E. coli or ST258 in Klebsiella) would be a great contribution for many phylodynamic studies which explicitly try to estimate parameters of bacterial evolution, or for the studies of the evolution of bacterial accessory genomes. 

Minor comments:

> Based on Figure 2 there seems to be a correlation between species abundance and the number of AMR genes. Is that driven by the clinical interest (problematic strains are more likely to be sequenced) or by a stronger signal to find AMR with enough genomes?

> It is unclear whether there is any chance of linking non-genetic metadata (eg, ecology or sampling date) with the ENA strains. If not (as I expect), it should be mentioned as one drawback of using a resource like this.

> I would really love to see a few basic summary plots of the assembly statistics, like the number of contigs, N50, CheckM parameters etc.), as one of the supplementary figures

> l. 116-117: this sentence is unclear; not ideal for what exactly, species identification?

> l. 542: I found y-axis title sub-optimal, maybe remove "covered"?

> Figure 1: some colours are hard to distinguish (eg, C. coli and C. difficile), I'd suggest changing the colour scale or showing genera, not species.

> Figure 2: please mark the most abundant species (say top 20). Also, why Actinobacteria are not shown? Please explain in the legend.

---

## [Editor Report · Decision Letter 2]

12 Sep 2021

Dear Dr. Blackwell,

Thank you for submitting your revised Methods and Resources entitled "Exploring bacterial diversity via a curated and searchable snapshot of archived DNA sequences" for publication in PLOS Biology. I have now discussed your revision with the Academic Editor. 

We will probably accept this manuscript for publication, provided you satisfactorily address the following data and other policy-related requests.

DATA POLICY:

Regardless of the method selected, please ensure that you provide the individual numerical values that underlie the summary data displayed in the following figure panels as they are essential for readers to assess your analysis and to reproduce it: Figures 2AB, 3ABC, Supplementary Figures 1ABCD, 2ABCD, 3, 4AB, 5AC, 6.

**Please also ensure that figure legends in your manuscript include information on where the underlying data can be found, and ensure your supplemental data file/s has a legend.**

We expect to receive your revised manuscript within two weeks.

*Published Peer Review History*

*Early Version*

Sincerely,

Paula

---

Associate Editor,

pjaureguionieva@plos.org,

PLOS Biology

---

## [Editor Report · Decision Letter 3]

20 Sep 2021

Dear Dr Blackwell,

I'm handling your manuscript on behalf of my colleague Dr Jauregui, who is out of the office for two weeks. I note that you have re-submitted your paper, BUT on looking at the file inventory, it seems that you may have forgotten to upload the latest versions of your files. My understanding is that Dr Jauregui sent the decision letter on Sept 12th, but the uploaded files all date from Aug 31st or before, and therefore don't contain the requested changes.

Please look at Dr Jauregui's previous decision letter for all the details, but essentially her sole requests were that you:

a) provide the underlying numerical values for Figures 2AB, 3ABC, S1ABCD, S2ABCD, S3, S4AB, S5AC, S6 (either as supplementary data files or as depositions in Figshare, Dryad, Github, etc.).

b) cite the location of the data clearly in each relevant main or supplementary Fig legend (e.g. "the data underlying this Figure may be found in S1 Data" or "the data underlying this Figure may be found in https://github.com/...."

Sincerely,

Roli Roberts

Roland G Roberts PhD

Senior Editor

PLOS Biology

rroberts@plos.org

on behalf of

Editor,

pjaureguionieva@plos.org,

PLOS Biology

---

## [Editor Report · Decision Letter 4]

21 Sep 2021

Dear Grace,

On behalf of my colleagues and the Academic Editor, William Hanage, I'm pleased to say that we can in principle offer to publish your Methods and Resources "Exploring bacterial diversity via a curated and searchable snapshot of archived DNA sequences" in PLOS Biology, provided you address any remaining formatting and reporting issues. These will be detailed in an email that will follow this letter and that you will usually receive within 2-3 business days, during which time no action is required from you. Please note that we will not be able to formally accept your manuscript and schedule it for publication until you have made the required changes.

PRESS: We frequently collaborate with press offices. If your institution or institutions have a press office, please notify them about your upcoming paper at this point, to enable them to help maximise its impact. If the press office is planning to promote your findings, we would be grateful if they could coordinate with biologypress@plos.org. If you have not yet opted out of the early version process, we ask that you notify us immediately of any press plans so that we may do so on your behalf.

Sincerely, 

Roli

Roland G Roberts PhD

Senior Editor

PLOS Biology

on behalf of

Paula Jauregui, PhD 

Senior Editor 

PLOS Biology
